# On the variability of the Bering Sea Cold Pool and implications for the biophysical environment

**Jaclyn Clement Kinney** [1] *, **Wieslaw Maslowski**[1], **Robert Osinski** [2], **Younjoo J. Lee**[1], **Christina Goethel**[3], **Karen Frey** [4], **Anthony Craig**[5]

**1** Naval Postgraduate School, Monterey, California, United States of America, **2** Institute of Oceanology, Polish Academy of Sciences, Sopot, Poland, **3** Chesapeake Biological Laboratory, University of Maryland Center for Environmental Science, Solomons, Maryland, United States of America, **4** Graduate School of Geography, Clark University, Worcester, Massachusetts, United States of America, **5** University of Colorado, Boulder, Colorado, United States of America

* jlclemen@nps.edu

**Data Availability Statement:** Model output that was used to generate the figures and analysis in this article can be obtained from the following

## Abstract

The Bering Sea experiences a seasonal sea ice cover, which is important to the biophysical environment found there. A pool of cold bottom water (<2˚C) is formed on the shelf each winter as a result of cooling and vertical mixing due to brine rejection during the predominately local sea ice growth. The extent and distribution of this Cold Pool (CP) is largely controlled by the winter extent of sea ice in the Bering Sea, which can vary considerably and recently has been much lower than average. The cold bottom water of the CP is important for food security because it delineates the boundary between arctic and subarctic demersal fish species. A northward retreat of the CP will likely be associated with migration of subarctic species toward the Chukchi Sea. We use the fully-coupled Regional Arctic System Model (RASM) to examine variability of the extent and distribution of the CP and its relation to change in the sea ice cover in the Bering Sea during the period 1980–2018. RASM results confirm the direct correlation between the extent of sea ice and the CP and show a smaller CP as a consequence of realistically simulated recent declines of the sea ice cover in the Bering Sea. In fact, the area of the CP was found to be only 31% of the long-term mean in July of 2018. In addition, we also find that a low ice year is followed by a later diatom bloom, while a heavy ice year is followed by an early diatom bloom. Finally, the RASM probabilistic intra-annual forecast capability is reviewed, based on 31-member ensembles for 2019–2021, for its potential use for prediction of the winter sea ice cover and the subsequent summer CP area in the Bering Sea.

## Introduction

The Cold Pool (CP) is the region of the Bering Sea shelf where bottom water is $< 2$˚C throughout the summer [1]. Cooling and seasonal sea ice formation in winter results in the formation of this cold, salty and dense water mass [1]. The freezing of surface water across the northern

website https://nps.box.com/s/ug6be32gdu7hpm49a5jarmimikq71xzr.

**Funding:** This work was supported by the US National Science Foundation (GEO/PLR ARCSS IAA1417888 and IAA1603602), the US Department of Energy (DOE) Regional and Global Model Analysis (RGMA) (89243019SSC0036 and DESC0014117), and the Office of Naval Research (ONR) Arctic and Global Prediction (AGP) (N0001418WX00364). The Department of Defense (DOD) High Performance Computer Modernization Program (HPCMP) provided computer resources.

**Competing interests:** The authors have declared that no competing interests exist.

Bering Sea begins in late autumn and continues, with the sea ice edge advancing further south, over the course of the winter. Sea ice production can be enhanced in polynya regions such as the St. Lawrence Island polynya [2, 3] and Sireniki Polynya [2]. (located south of the Chukotka Peninsula), as well as in other smaller polynyas.

The cooling in autumn followed by salinization of the surface waters due to sea ice formation and subsequent brine rejection in winter leads to overturning and vertical mixing throughout the water column. This allows for the formation of a bottom water CP. Later in the year, during spring and summer, as the surface water is warmed by solar insolation and advection, a seasonal pycnocline develops, which isolates and protects the bottom water of the CP [1].

The maximum sea ice extent in the Bering Sea is typically reached in March and can extend as far south as the shelf break. However, the maximal extent can vary by up to hundreds of kilometers each year. Important factors influencing this variability include: variation in the wind field, due to changes in the strength and positions of the Aleutian Low and Siberian High, northward oceanic heat transport, as well as variation in air and sea surface temperatures [4]. The persistence of seasonal sea ice and time of melt is interannually variable, as well, which impacts the availability of light and stratification of the water column. These factors directly affect the ecosystem of the Bering Sea [5, 6].

All of this variability in sea ice affects the distribution of the CP. Years with heavy ice production and a high sea ice extent produce an extensive CP that may reach far south into Bristol Bay. While years with little ice production yield a limited CP that is below average and may not ever reach down to St. Matthew Island [1]. For example, recent observations by Duffy-Anderson et al. [7] revealed an absence of the summer CP in 2018 within their observational domain covering the central and eastern Bering Sea, following a winter of anomalously low sea ice extent.

The CP is important for ecosystem structure and delineates the boundary between Arctic and subarctic fish species [8]. Bottom water temperature has been found to be the dominant climate parameter for determining community composition of demersal fish species and some pelagic species in the Bering Sea [9]. Arctic cod tend to be found within the CP, while walleye pollock prefer warmer temperatures and tend to stay outside the boundary of the CP [1]. Changes in the distribution of various species due to variability in the size and location of the CP have been noted [1, 9, 10]. The Bering Sea contains important stocks of commercial fish species hence understanding the changing physical environment in which they live is critical for proper management practices. For example, Bering Sea snow crab have been shown to retreat northward with a retreating CP distribution [9]. Additionally, the exclusion of predatory demersal fish promotes the rich and high benthic biomass [11]. that lives within the boundary of the CP in the northern Bering Sea [12]. that are prey for upper trophic benthivoure predators, such as spectacled eiders, Pacific walrus, and bearded seals (i.e. [11, 13, 14]).

Earlier model simulations of the Bering Sea CP include those of Zhang et al. [15]. who showed that increased stratification, both from sea ice melt and surface heating, isolates the CP from surface effects. They also noted that the isolation effect was greater in cold years with more sea ice in winter and, subsequently, more melting. More recently, Kearney et al. [16]. compared results from the coarse-resolution Climate Forecast System model with those from a higher-resolution model. They found that the dynamic downscaling offered by the higher-resolution model was essential for reproducing the cold bottom water temperatures of the CP. Near-bottom temperature observations on the southeastern Bering Sea shelf have noted changes in the distribution of the CP [e.g. [7, 17]], especially in the central and eastern parts of the shelf. We present here a long-term timeseries of the full size of the CP, including its extension into Russian waters, utilizing results from the Regional Arctic System Model (RASM;

[18]) and examine the linkage between CP area and sea ice area. The distribution of sea ice is shown to exert a strong control on the size of the CP and it is also found to have an important effect on the timing of phytoplankton blooms on the shelf. We also analyze results from the RASM ensemble forecasts of sea ice and bottom water temperature in the Bering Sea, for their potential usefulness to stakeholders in the region.

## Model description, methods, and validation

RASM has been developed to address some of the limitations of global Earth System models, including the representation of Arctic relevant processes within and coupling among model components and high spatio-temporal resolution. It is a fully-coupled regional Earth system model with components including, atmosphere, ocean, sea ice, marine biogeochemistry, land hydrology and a river routing scheme. All the components are coupled using the flux coupler of Craig et al. [19]. The RASM domain includes all the sea-ice covered ocean in the Northern Hemisphere, the Arctic river drainage, and large-scale atmospheric weather patterns. This domain includes all oceanic areas of the northern hemisphere from 90˚N to 55˚N and most of the North Pacific down to 30˚N. We use it here as an optimal tool to investigate the Cold Pool and related bio-physical processes in detail.

The RASM components and their configurations are described in Table 1. The ocean bio-geochemistry (BGC) component in RASM is a medium-complexity Nutrients-Phytoplankton-Zooplankton-Detritus (NPZD; [20]) model. The sea ice model uses a BGC parameterization, in which biological activity is distributed throughout the ice column [21]. RASM has been in development by a dedicated team for over a decade. Each component of this system model, as well as the fully-coupled system, has been validated and simulation results evaluated extensively [22–30].

The RASM historical (1979–2018) simulation results analyzed here were produced after a 78-year spinup, which started with no sea-ice and the Polar Science Center Hydrographic Climatology (PHC) 3.0 [31] climatological ocean temperature and salinity at rest and was forced with the Coordinated Ocean-sea ice Reference Experiments version 2 (CORE2) reanalysis [32]. The RASM 6-month 31-member ensemble forecast results presented here were initialized on February 1 of each forecast year. The forcing along the lateral boundaries and in the upper atmosphere, as well as atmospheric initial conditions, were derived from the National Centers for Environmental Predictions (NCEP) global Coupled Forecast System (CFS) version 2 (CFSv2) operational 9-month forecasts initialized at 0000Z each day of the preceding month. The ocean and sea ice initial conditions for these forecasts have been produced from a separate fully-coupled RASM hindcast simulation (without BGC components), which started in September 1979 and has been updated through January 2021 using surface atmospheric forcing from the NCEP global CFS Reanalysis (CFSR).

**Table 1. Components, code and configuration of RASM.**

| Component | Code | Configuration |
| --- | --- | --- |
| Atmosphere | WRF3 | 50km, 40 levels |
| Land | VIC | 50km, 3 soil layers |
| Ocean + Biogeochemistry | POP2 | 1/12˚(~9km), 45 levels |
| Sea Ice + Biogeochemistry | CICE6 | 1/12˚(~9km), 5 thickness categories |
| Coupler | CPL7x | Flux exchanges every 20 min |

The code abbreviations above include: Weather Research and Forecasting version 3 (WRF3) atmosphere model, the Variable Infiltration Capacity (VIC) land hydrology model, the Los Alamos National Laboratory (LANL) Parallel Ocean Program version 2 (POP2) and Sea Ice Model version 6 (CICE6), and the Flux Coupler (CPL7x).

The sea ice representation by RASM has been recently evaluated and validated by Watts et al. [33]. This work utilized spatial analysis metrics, such as the integrated ice edge error, Brier score, and spatial probability score to examine the historical representation (1980–2014) of sea ice extent, volume, and thickness. Results from RASM, as well as 12 other models which participated in phase 6 of the Coupled Model Intercomparison Project (CMIP6), were compared with the available satellite observations of sea ice parameters. Of the 13 models that were examined, RASM had the best spatial probability score and lowest integrated ice edge error for representation of the Bering Sea ice extent during the months of June to November (see Table 3 of Watts et al. 2021 and S2 of Watts et al. 2021). RASM also had the best representation of the March Bering Sea ice extent (see Figs 8 and 10 of Watts et al. 2021), with an underestimation of approximately 10,000 km$^2$ (or ~2.5% of the total) in the mean.

Results from the predecessor to this present configuration of RASM BGC was reported on in Jin et al. [28]. This work found that moving to a higher spatial resolution and adding an improved mixed layer depth scheme improved simulation of nitrate concentrations and primary production, especially in areas with sharp bathymetric gradients such as along shelf breaks in the Western Arctic. Their 9-km model configuration (R9km), which produced results with the lowest biases in the Jin et al. [28] study, is the predecessor to the updated model version used and analyzed here. Some of the key updates we have made include upgrading CICE to version 6, as well as utilizing the fully-coupled configuration of RASM.

More recently, results from the current BGC component of RASM have been examined by Frants et al. [34] and Clement Kinney et al. [24]. In the Frants et al. [34] study, it was found that RASM realistically represents the phytoplankton bloom of pelagic diatoms found under sea ice in a similar place and time to observations in the Chukchi Sea from 2011. However, the timing of the modeled bloom was approximately 2 weeks earlier, due to an earlier retreat of the sea ice over the northern Chukchi Sea, likely caused by discrepancies in the predicted wind field over the region. In addition, the modeled values of chl-*a* underestimated the *in situ* measurements, especially in terms of the maximum chl-*a* concentrations, likely due to the under-representation of the mesoscale physical processes, e.g. eddies or vertical mixing along the ice edge, and associated nutrient concentrations potentially captured by point measurements. The recent Clement Kinney et al. [24] study was motivated by limited observations of phytoplankton blooms beneath Arctic sea ice and the lack Arctic-wide primary production observational estimates beneath the ice. Using the fully-coupled RASM with marine biogeochemistry it was estimated that 63%/41% of the total primary production in the Arctic occurs in waters covered by sea ice that is ≥50%/≥85% concentration. This means that the majority of total primary production in the Arctic may be missing from current observational estimates based on satellite data. The total primary production in the Arctic was found to be increasing at a rate of 5.2% per decade during 1980–2018. Increased light transmission, due to the removal of sea ice, more extensive melt ponds, and thinner sea ice, was implicated as the main cause of increasing trends in primary production.

## Results

The Bering Sea bathymetry, relevant place names, and area for CP calculation are shown in Fig 1 for reference. A timeseries of the monthly mean CP area based on model results from 1980–2018 is shown in Fig 2. The area for calculation of the CP extends from 55-66˚N latitude and across all longitudes of the Bering Sea; deep water (> 250 m) is excluded (Fig 1 shows the area in pink contours). A strong annual cycle is present in the timeseries, as well as interannual variability. In particular, there is a clear increasing trend in the 2000s and a decreasing trend since 2012. The mean CP area annual cycle (over 1980–2018) is shown in Fig 3 with a maximum in

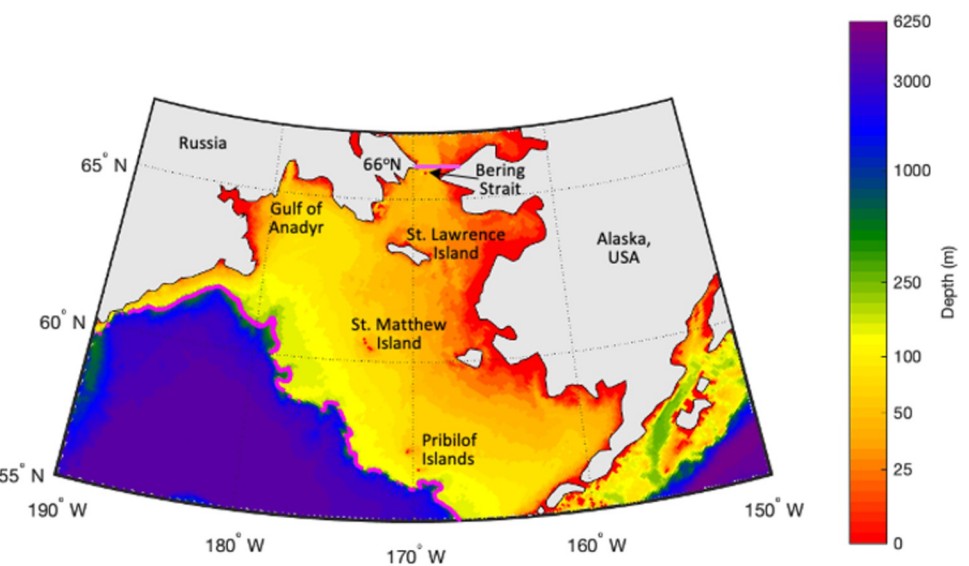

**Fig 1. Bathymetry and region of interest.** Bathymetry (m) of the Bering Sea, relevant place names, and the area for cold pool calculation (pink contours). Figure was created from the authors' data.

March (691,000 km$^2$) and a minimum in October (180,000 km$^2$). Because recent CP observational work [7] focuses on the summer bottom water temperatures, we will also focus on the July CP values. The annual cycle of individual years is shown as black lines with a large range

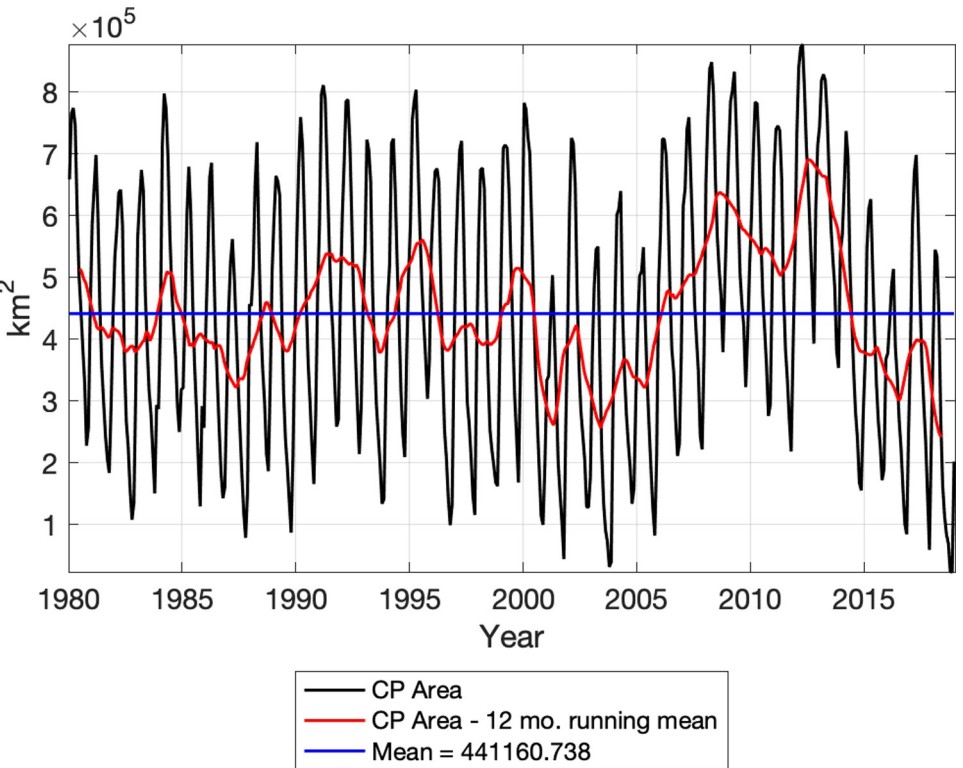

**Fig 2. Monthly mean Cold Pool area.** Cold Pool area (km$^2$; black), 12-month running mean of the Cold Pool area (km$^2$; red) and the long-term mean (km$^2$; blue) from RASM during 1980–2018.

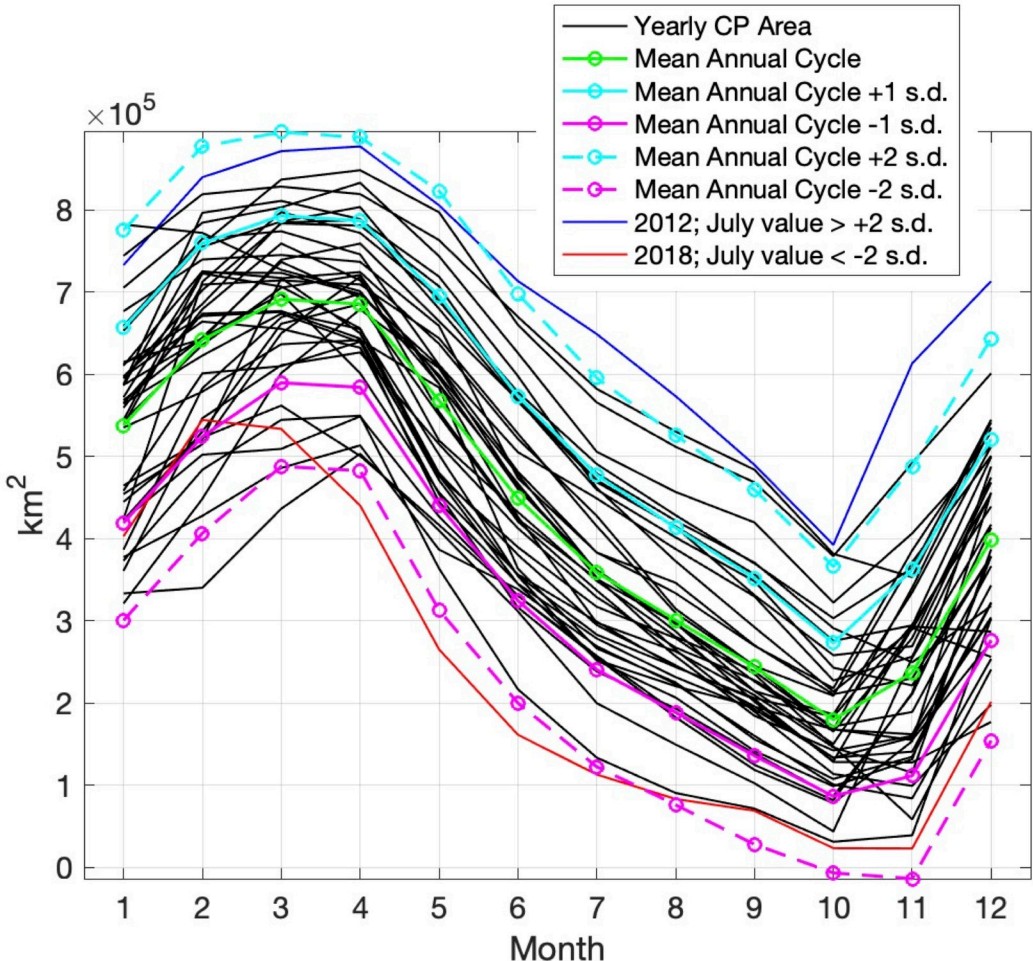

**Fig 3. Annual cycle of the Cold Pool area.** Annual cycle of the Cold Pool area (km$^2$) from RASM based on the years 1980–2018 (black) and long-term mean (1980–2018; green). Cyan and magenta lines indicate +/- 1 (solid) and 2 (dashed) standard deviations. The years 2012 and 2018 (blue and red, respectively) have July values outside 2 standard deviations from the mean.

of variability (1 standard deviation during July is +/- 118,000 km$^2$). Two years exhibit July CP area values outside of 2 standard deviations; these are 2012 (649,000 km$^2$ or 290,000 km$^2$ above the mean) and 2018 (113,000 km$^2$ or 246,000 km$^2$ below the mean).

Fig 4 shows the monthly mean July CP area from 1980–2018. Overall, the July CP area timeseries shows lower variability around the mean during the 1980s and 1990s as compared with the higher variability in the 2000s. The CP area was below average during the early 2000s, followed by a strong increase in the late 2000s and finally a steep decline following the maximum in 2012. The dramatically low, minimal value in July 2018 (113,000 km$^2$) was only 31% of the long-term mean July area (359,000 km$^2$). Two other minima (2001 and 2003) are noted in the July timeseries, with a July 2001 value of 200,000 km$^2$ and a July 2003 value of 134,000 km$^2$. These minima can be contrasted with the maximum of the timeseries in July 2012 (649,000 km$^2$ or 181% of the mean). The colored circles in Fig 4 relate to the March sea ice conditions for those years and are discussed later.

Fig 5 gives a spatial view of the bottom water temperature in July for the long-term mean (1980–2018), 2012 (year of maximum CP area), and 2018 (year of minimum CP area). The CP

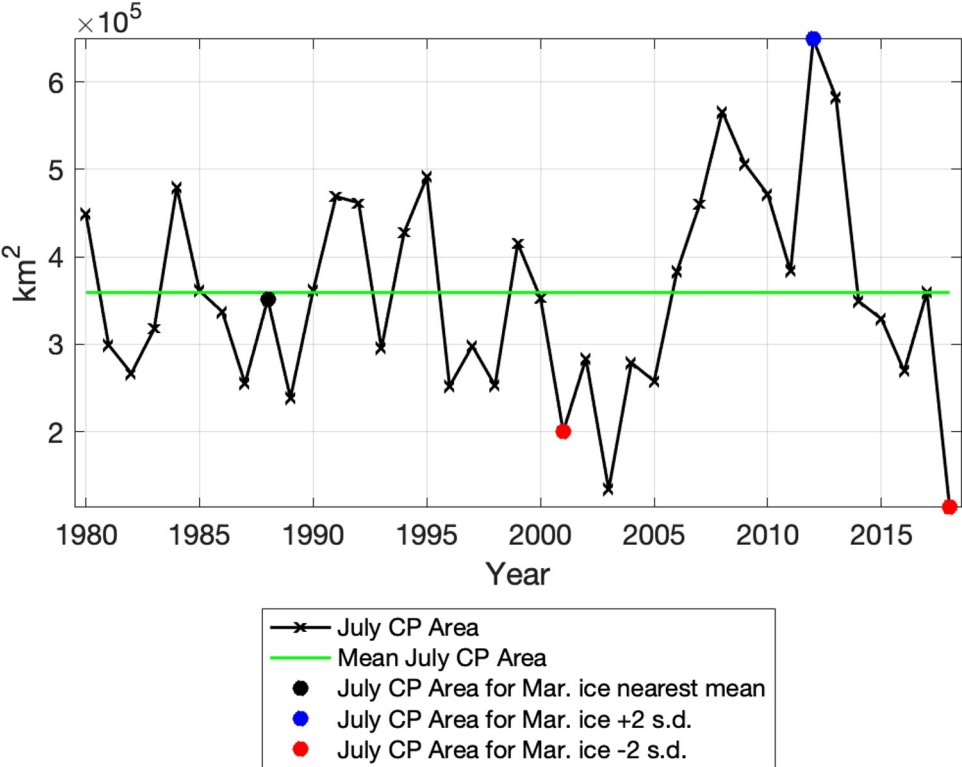

**Fig 4. July CP area.** July CP area (km$^2$) from RASM for 1980–2018 (black line). Mean July (1980–2018) CP area (km$^2$) green line. The black circle indicates the year (1988) when March ice area was nearest the long-term mean. The blue circle indicates the year (2012) with March ice area 2 standard deviations (s.d.) above the mean. The red circles indicate the years (2001 and 2018) with March ice area 2 s.d. below the mean.

area (indicated by the black line) shows that the CP typically covers almost the entire Gulf of Anadyr and most of the central shelf with a smaller tongue reaching south-eastward past 170˚W. During 2012, the CP area reaches all the way into Bristol Bay past 160˚W and covers almost the entire shelf, except for areas of coastal Alaska. In contrast, the CP in 2018 is extremely restricted, with a presence only in the Gulf of Anadyr and no bottom water <2˚C on the central or eastern shelf. In fact, the bottom water positive temperature anomalies in RASM were in excess of 4˚C south of St. Lawrence Island during July 2018.

Since sea ice formation has an important effect on the temperature of the water column and the formation of the CP [1, 4, 7, 15] we analyze the Bering Sea ice area next. Fig 6 shows a mean (1980–2018) monthly climatology of sea ice concentrations across the Bering Sea region, calculated using data from the Scanning Multi-channel Microwave Radiometer (SMMR), Special Sensor Microwave/Imager (SSM/I), and Special Sensor Microwave Imager/Sounder (SSMIS) passive microwave instruments, and utilizing the Goddard Bootstrap (SB2) algorithm [35, 36]. Seasonal sea ice across the region is most extensive in March and pulls back to minimal concentrations during June through October. November and December are typically when seasonal sea ice begins to reform, growing southward with each progressive month through the winter. A timeseries of Bering Sea ice area from RASM is shown in Fig 7. The strong annual cycle of sea ice area is present with the melting of all sea ice each summer. The years of 2001, 2015, 2016, and 2018 stand out in the timeseries, with wintertime ice areas less than 400,000 km$^2$, both in the model and in the satellite observations. The model underrepresents the wintertime peak of sea ice area observed by satellite. The mean annual cycle of sea ice

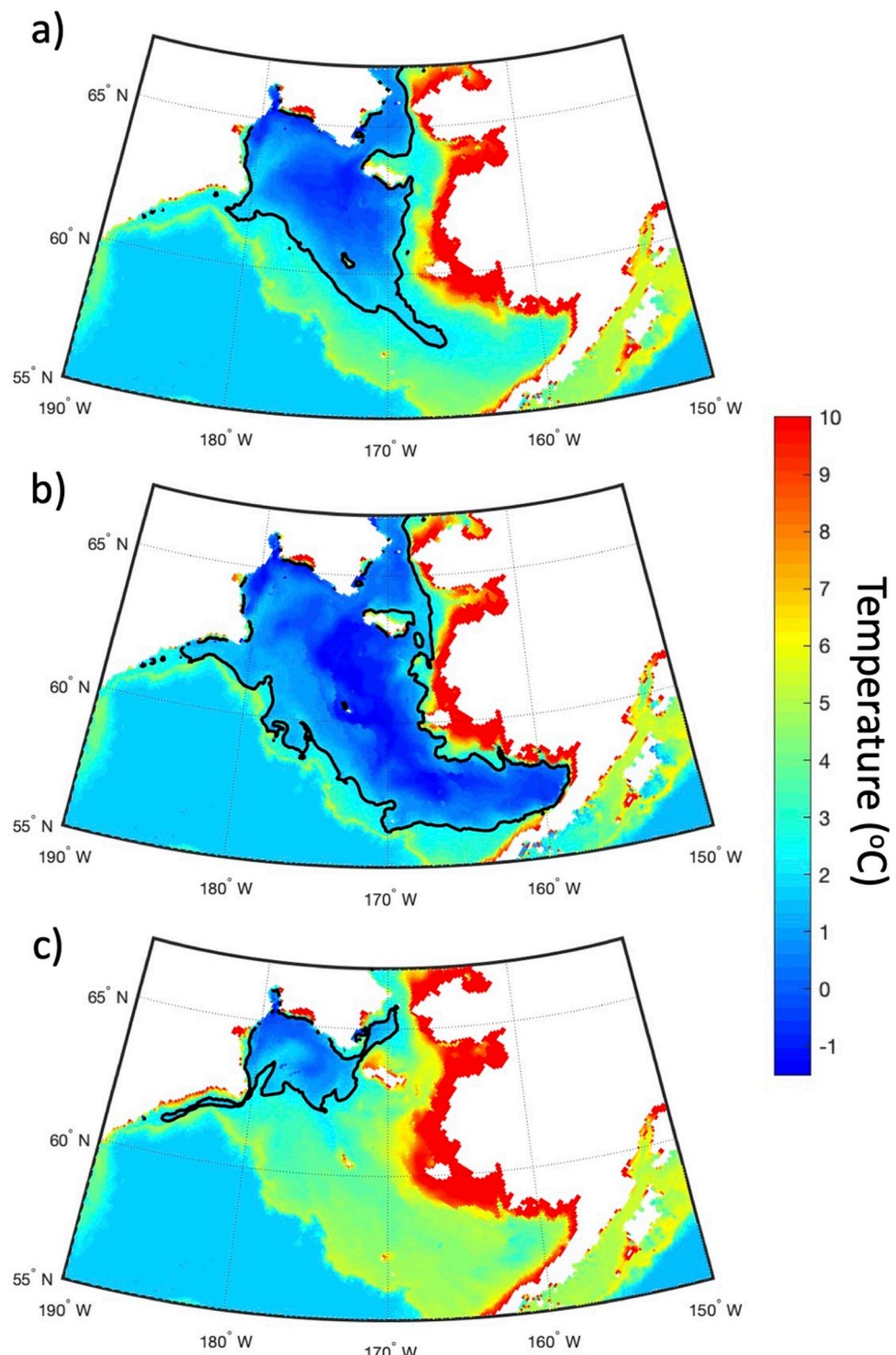

**Fig 5. Bottom water temperature.** Bottom water temperature (°C) during July 1980–2018 mean (a), 2012 (b), and 2018 (c) as calculated by RASM. The black contour line indicates the position of the 2°C isotherm on the Bering Sea shelf and is the boundary of the CP. Figure created using Matlab (v. 2018a) from the authors' data.

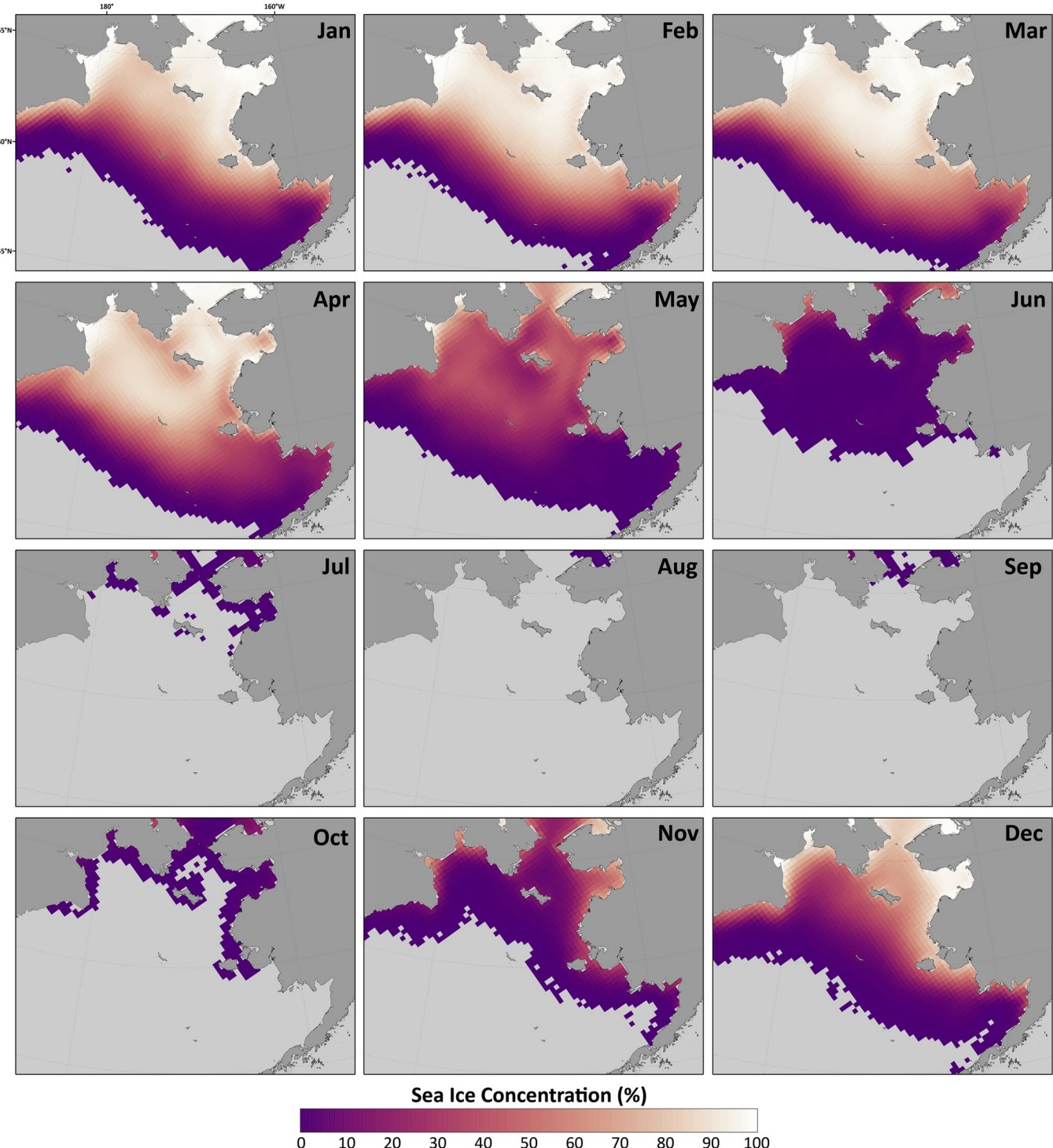

**Fig 6. Mean monthly sea ice concentration (1980–2018).** Mean sea ice concentration from the Scanning Multi-channel Microwave Radiometer (SMMR), Special Sensor Microwave/Imager (SSM/I), and Special Sensor Microwave Imager/Sounder (SSMIS) passive microwave instruments, calculated using the Goddard Bootstrap (SB2) algorithm [35, 36]. Figure created using Matlab (v. 2018a) from the authors' data.

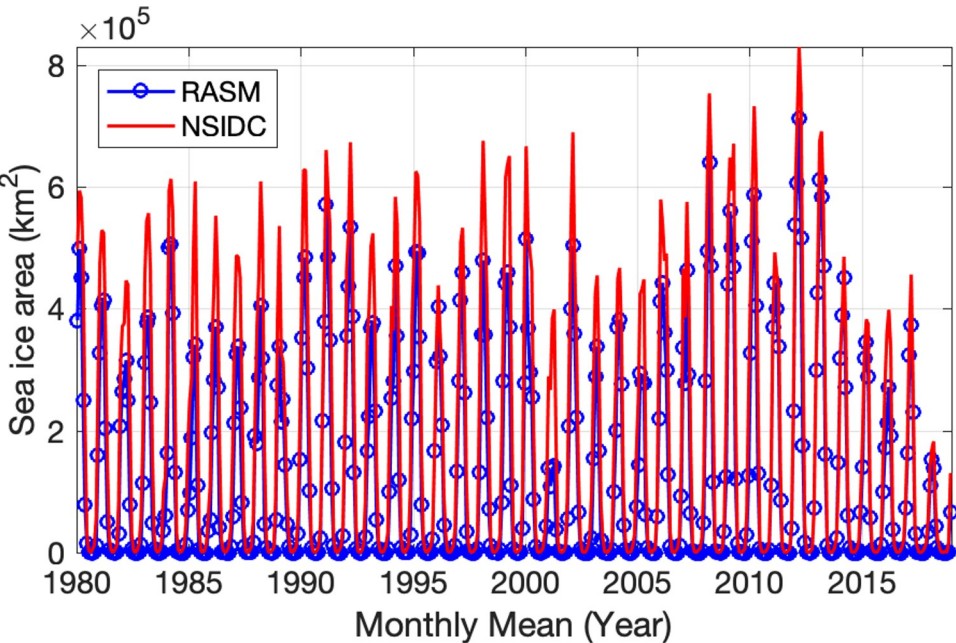

**Fig 7. Monthly mean sea ice area.** Monthly mean sea ice area (km$^2$) in the Bering Sea calculated from RASM (blue) and the National Snow and Ice Data Center (NSIDC; red) satellite product [37].

area in the Bering Sea from RASM is shown in Fig 8 (green line). Ice area reaches a peak in March (mean of 406,000 km$^2$) and then begins to melt in April. Sea ice is typically low to absent in the Bering Sea from June thru October and begins to form again in November. The satellite-observed mean annual cycle (orange line) is very similar to RASM, however RASM has a low bias throughout the year. Fig 8 also shows the interannual variability in sea ice area (black lines) with a March s.d. of +/- 120,000 km$^2$. Similar to observations [7] RASM indicates that, during the period of satellite data, 2012 was the year of maximum sea ice area (713,000 km$^2$) and 2018 was the minimum (139,000 km$^2$). RASM also shows another minima during 2001 that is also less than 2 s.d. below the mean. This minima was associated with the third lowest July CP area (Fig 4) and is discussed further in the Discussion.

Monthly mean values of ice area and CP area (over the years 1980–2018) show a significant (p-value < 0.01) correlation with a value of 0.79. Even after removing the annual cycle, the correlation is still significant with a value of 0.60. Fig 9 shows a spatial view of the sea ice area during April (monthly mean) for the long-term mean, 2012 (maximum year), and 2018 (minimum year). Fig 9A shows that the long-term mean ice coverage extends all the way south to St. Matthew Island with some low concentrations south of 60°N. The maximum distribution (Fig 9B) has values > 80% for most of the western and central shelf extending south to the shelfbreak, except in polynya regions (e.g. the St. Lawrence Island Polynya and Sireniki Polynya) and the southeast Bering where concentrations are lower (20–50%). However, during 2018 (Fig 9C) ice was present only in coastal areas of the Gulf of Anadyr and other areas north of St. Lawrence Island. Compared to satellite observations (green line in Fig 9B and 9C), RASM is representing the spatial distribution very well, but the extent does not reach as far south as the data from NSIDC.

Modeled primary production is strongly affected by the presence or absence of sea ice and the timing of its formation and melt. Fig 10 shows the RASM-simulated surface primary production during the peak months of May and June for 2012 and 2018. May 2012 shows a large

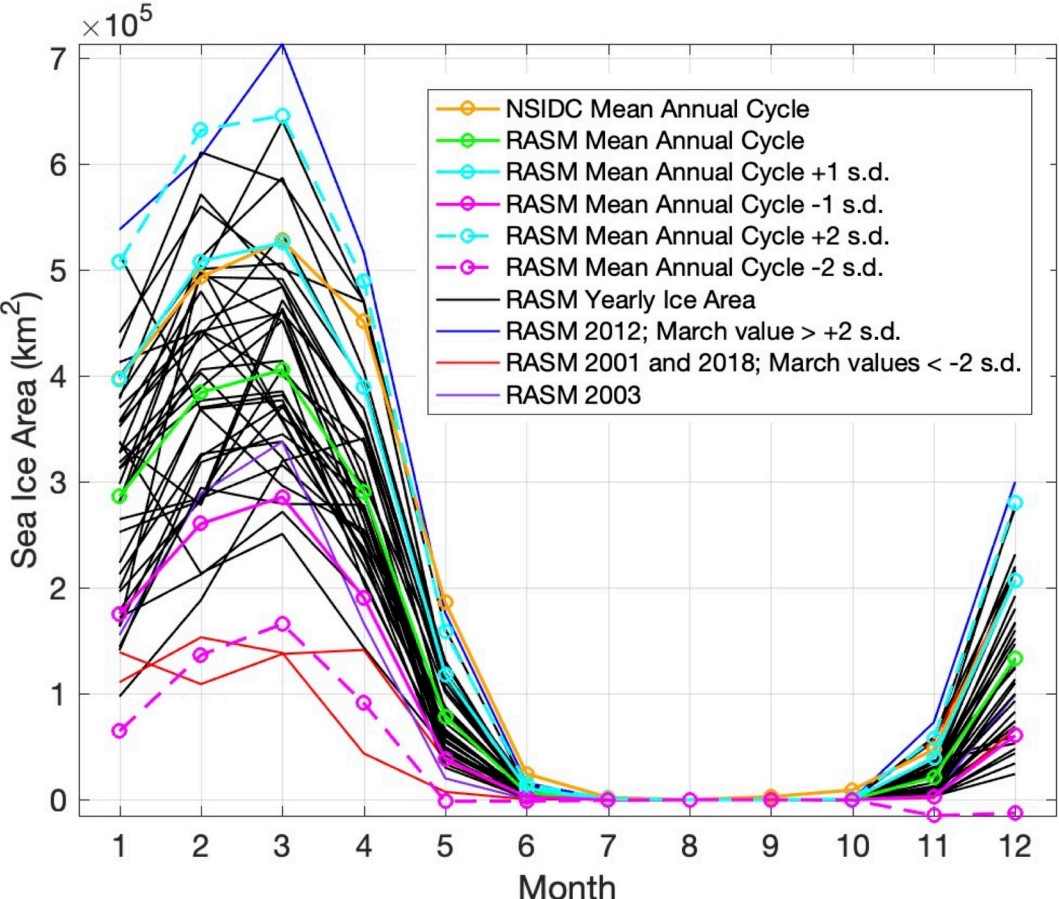

**Fig 8. Annual cycle of sea ice area in the Bering Sea.** Annual cycle of sea ice area in the Bering Sea (km$^2$) from the National Snow and Ice Data Center (NSIDC; orange) satellite product [37] (long-term mean based on the years 1980–2018), from RASM (long-term mean based on the years 1980–2018; green), and yearly annual cycles from RASM (black). Cyan and magenta lines indicate +/- 1 (solid) and 2 (dashed) standard deviations. The year 2012 (blue) and years 2001 and 2018 (red) have March values outside 2 standard deviations from the mean. The year 2003 is shown in purple.

bloom with values in excess of 100 mg C / m$^3$ / day, whereas June 2012 shows a dwindling bloom. This can be contrasted with 2018 where May has weak values on the central and southeastern shelf, but considerably higher production during June. Vertical profiles at 60.5˚N, 175˚W (location is shown as small black circles in Fig 10) of temperature, salinity, and density from RASM reveal differences in the stratification of the water column during these two years (Fig 11). Beginning with May, there is stratification of the water column in 2012 due to salinity because of sea ice melt; however, during 2018 there was very little sea ice formation/melt and therefore no stratification of the water column. Moving on to June, we see the development of thermal stratification during both 2012 and 2018. The thermal stratification in June 2018 allows for the development of the phytoplankton bloom seen in Fig 10. This June bloom in 2018 is later than the peak of the bloom in 2012, which occurred in May.

## Discussion

The extreme reduction in sea ice formation during 2018 resulted in the smallest CP area over the model timeseries from 1980–2018 (Fig 4). Modeled bottom water temperatures compare well with observations made by Duffy-Anderson et al. (2; see their Fig 1) and Goethel et al.

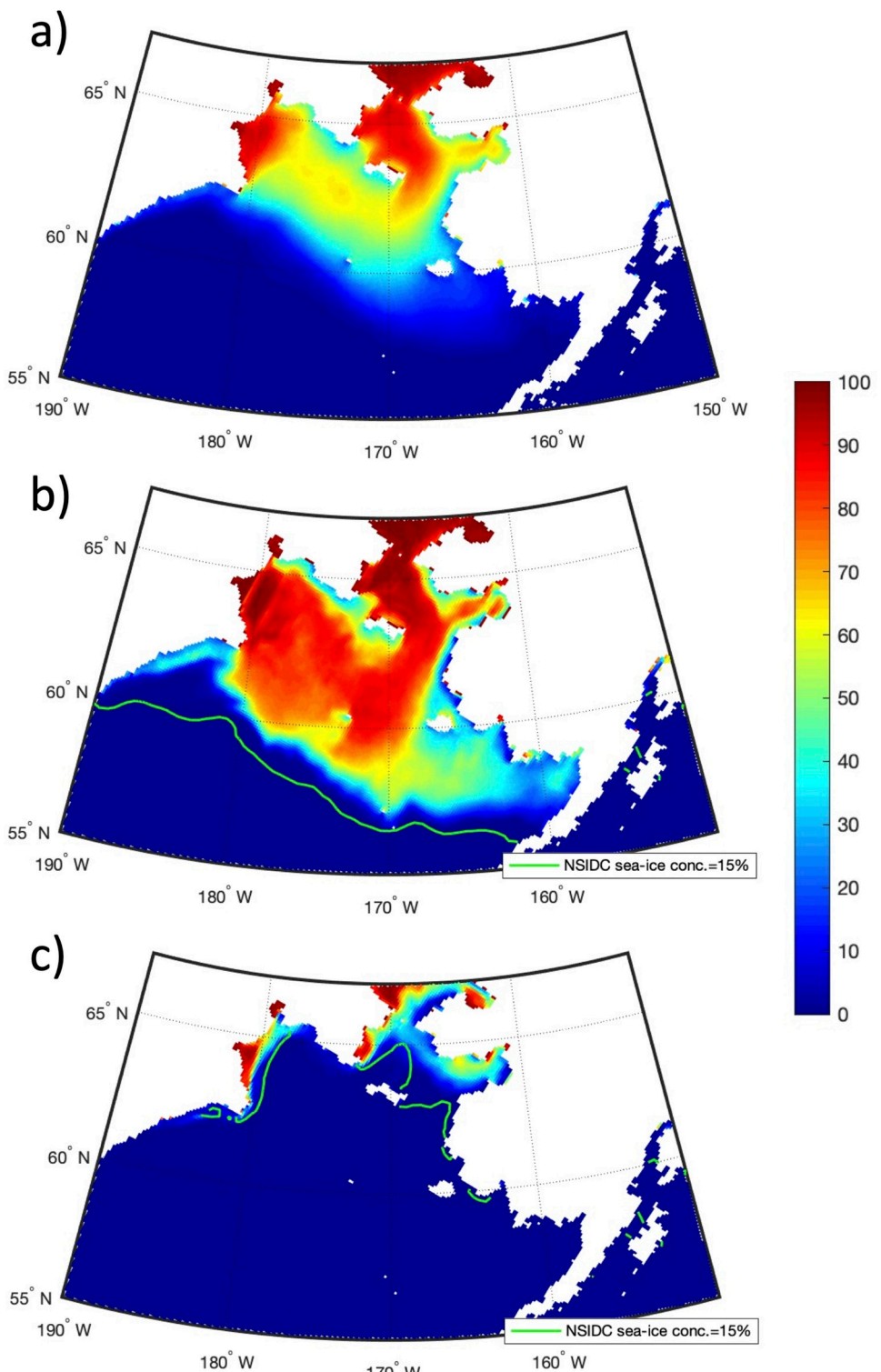

**Fig 9. April sea ice concentration.** April sea ice concentration (%) from RASM during 1980–2018 mean (a), 2012 (b), and 2018 (c). Observations from the National Snow and Ice Data Center satellite product [37] are shown as green lines for the 15% ice area contour. Figure created using Matlab (v. 2018a) from the authors' data.

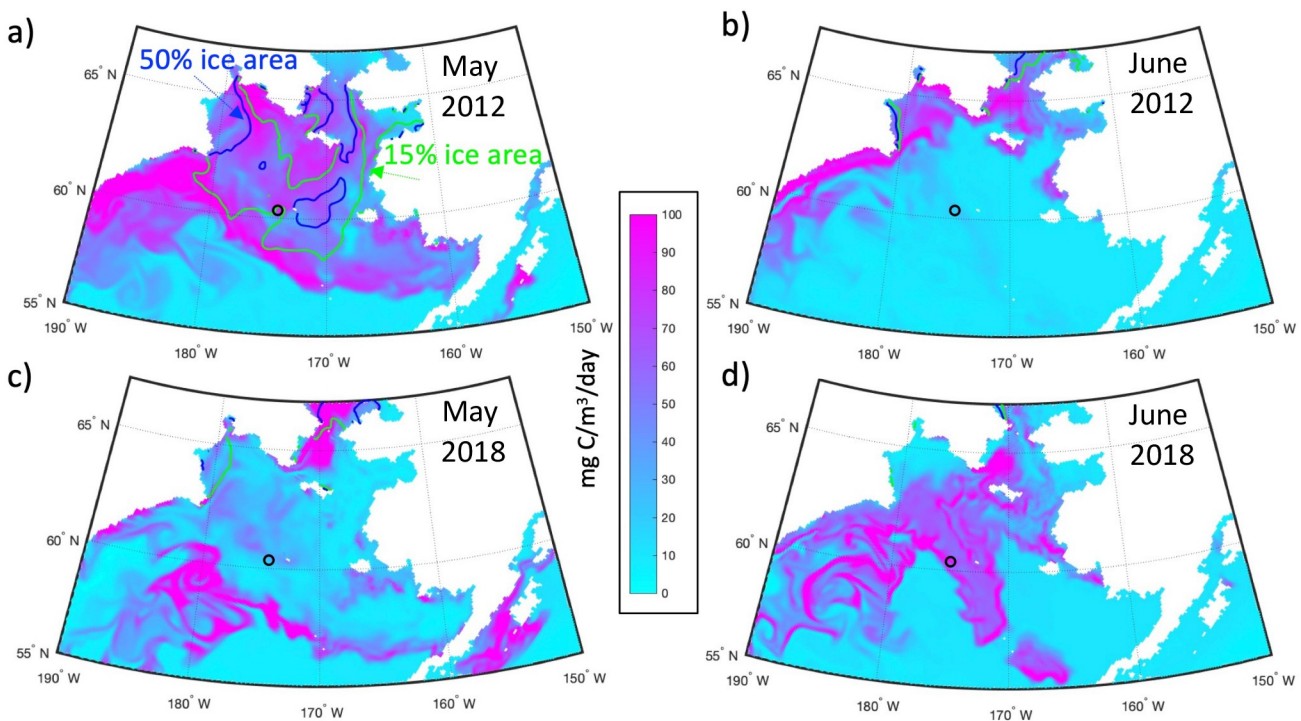

**Fig 10. Surface primary production.** Surface primary production (mg C / m³ / day) during May (a, b) and June (c, d) for years 2012 (a, c) and 2018 (b, d) from RASM. The blue line indicates the 50% ice area contour and the green line indicates the 15% ice area contour. Small black circles at 60.5˚N, 175˚W denote the location of vertical profiles shown in Fig 11. Figure created using Matlab (v. 2018a) from the authors' data.

2021 (this issue; see their Fig 1), which show the lack of a Cold Pool on the central and eastern shelf during summer 2018 (Fig 5). RASM has a realistic representation of the extremely low ice area for 2018, in comparison with the ice area observed by satellite (Special Sensor Microwave Imager / Sounder (SSMIS); Frey et al. 2021, this issue). The low ice area is largely due to the fact that the Bering Sea experienced strong winds out of the south in winter 2018 that restricted the typical southward expansion of sea ice toward the shelf break [7]. It is interesting to note that the winter sea ice minimum was reached only 6 years after the timeseries maximum in 2012, which reflects the strong dependence of the winter sea ice cover on the wind field in the Bering Sea.

We have focused much attention on the dramatically low CP and sea ice area during 2018. However, 2001 also saw a very small Cold Pool and sea ice area. Clement at al. (2005) reported on the lack of sea ice in winter 2001 and related it to the unusual winds that were present during November-December 2000. Strong winds out of the east reduced the typical sea ice growth and southward expansion and instead pushed ice toward the Gulf of Anadyr. The other relatively low CP minima in 2003 (Fig 4) was not preceded by such anomalously low sea ice area as in 2001. However, the sea ice melted much earlier than the mean in April and May 2003 (see purple line in Fig 8), which may have contributed to the reduced CP area later in the summer.

Interannual variability in sea ice presence affects not only the formation of the CP, but also the timing of spring phytoplankton blooms on the shelf. Variability in primary production appears to follow the Oscillating Control Hypothesis originally developed by Hunt et al. (2002) for the southeastern Bering Sea. The hypothesis states that early ice retreat will lead to a late bloom, while late ice retreat leads to an early bloom. During a heavy ice year, like 2012, the

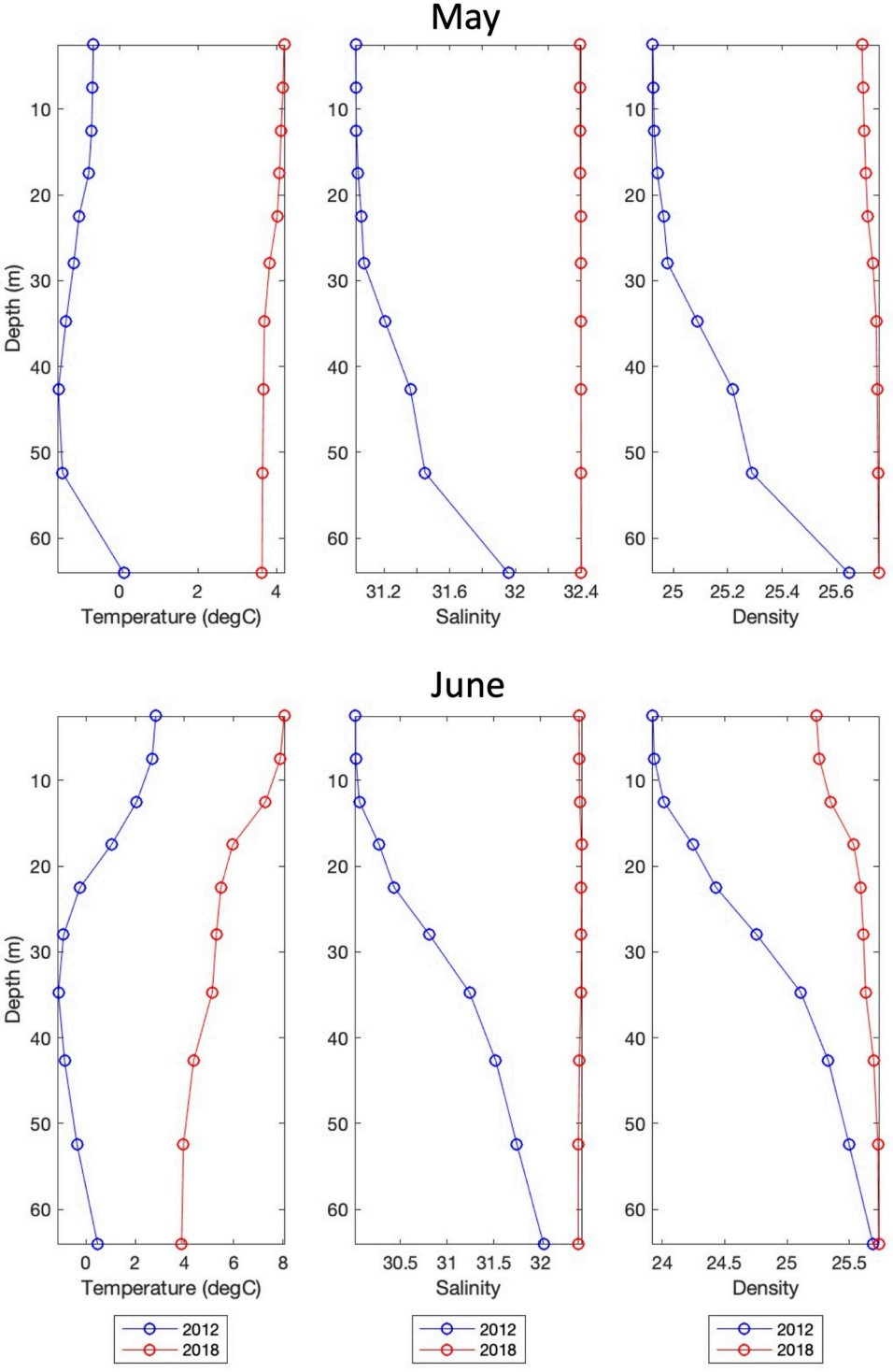

**Fig 11. Vertical sections of temperature, salinity, and density.** Vertical sections of temperature (˚C), salinity, and density (σ) during May (upper) and June (lower) for 2012 (blue) and 2018 (red) from RASM at 60.5˚N, 175˚W (location shown as a small black circle in Fig 10).

bloom happens immediately after sea ice melt in May when low surface salinity helped stratify the water column. On the other hand, during a light ice year, like 2018, the bloom across the central shelf did not occur until June after thermal stratification had developed. This has implications for the pelagic versus benthic ecosystems. In the current regime, where blooms occur early in the season, much of the production sinks to the bottom due to limited grazing by zooplankton. The sinking of this production helps to support a rich benthic community in the Bering Sea, as well as the rest of the Pacific Arctic. Food availability to the benthos is one of the largest drivers for both the benthic community as a whole [38, 39] and to specific species such as *Macoma calcarea*, a common bivalve in the area [40]. Bottom water temperature, in this case the presence of the CP, has also been shown as a physical driver for some benthic organism populations [40]. Therefore, years with extensive sea ice are expected to likely favor the benthos since much of the production will continue to sink to the bottom without being consumed and grazed upon by zooplankton. However, in years with little sea ice and warmer temperatures there could be a shift to a more pelagic community, as zooplankton growth, abundance, and grazing could increase [41], and zooplankton species that are advected into the system may be able to survive better in these warmer waters [42], thus weakening the notable pelagic-benthic coupling of the region.

Due to the high degree of interannual variability in sea ice area and CP area on the Bering Sea shelf, it would be useful to have forecasts of these fields skillful enough to provide stakeholders with an estimate of conditions for the upcoming months. Such an objective has motivated the development of RASM intra-annual (i.e., up to 6 months) ensemble forecasts. Every month since January 2019, RASM has been used to generate (28–31; depending on the number of days in the month) probabilistic forecasts of the sea ice conditions into the future, by dynamically downscaling the global CFSv2 operational 9-month forecasts (https://nps.edu/web/rasm/welcome). While these products are not meant for commercial or operational use, they provide useful and often more realistic insights on conditions at a regional to local scale, in addition to what is available from global forecasts. Fig 12A shows the prediction of sea ice concentration of 15% (i.e. sea ice extent) for April 15, 2019, which was initialized on February 1, 2019. The ensemble mean (based on 31 members) prediction of sea ice extent is shown with a purple contour line, while the light blue line in Fig 12A shows the RASM hindcast sea ice extent for the same date. The observed ice extent (green line) is in the middle of the RASM forecast and hindcast. The dark blue and red lines show the respective maximum and minimum sea ice extent in the Bering Sea, out of the 31 ensemble members. A large spread between the predicted maximum and minimum of sea ice extent based on individual forecasts exemplifies both a large variability of possible winter sea ice conditions in response to winds and a challenge in their prediction at sub-seasonal and longer time scales. A statistical analysis of the RASM forecasts is found in Figs 15–17 and discussion of these results is found below. Fig 12B shows the probability of sea ice extent (i.e. with concentrations ≥15%) on April 15, 2019, along with the observed sea ice extent (green) contour. In agreement with the observed ice extent, the highest probabilities were predicted on the northern part of the shelf, particularly in the Gulf of Anadyr and north of St. Lawrence Island.

Similarly, Fig 13A shows the prediction of sea ice extent for April 15, 2020, initialized on February 1, 2020. As in Fig 12A, a very large spread exists between the minimum and maximum ice extents predicted from the ensemble of 31 members. In both years the minimum ensemble member shows an ice edge north of St. Lawrence Island, while the maximum ensemble member shows ice approximating the shelf break in the western and central Bering Sea. Fig 13B shows the probability of sea ice area >15% on April 15, 2020 based on a 31-member ensemble forecast. RASM predicted more sea ice, with an extent further south, in 2020 than in 2019 and this was observed by NSIDC, as well.

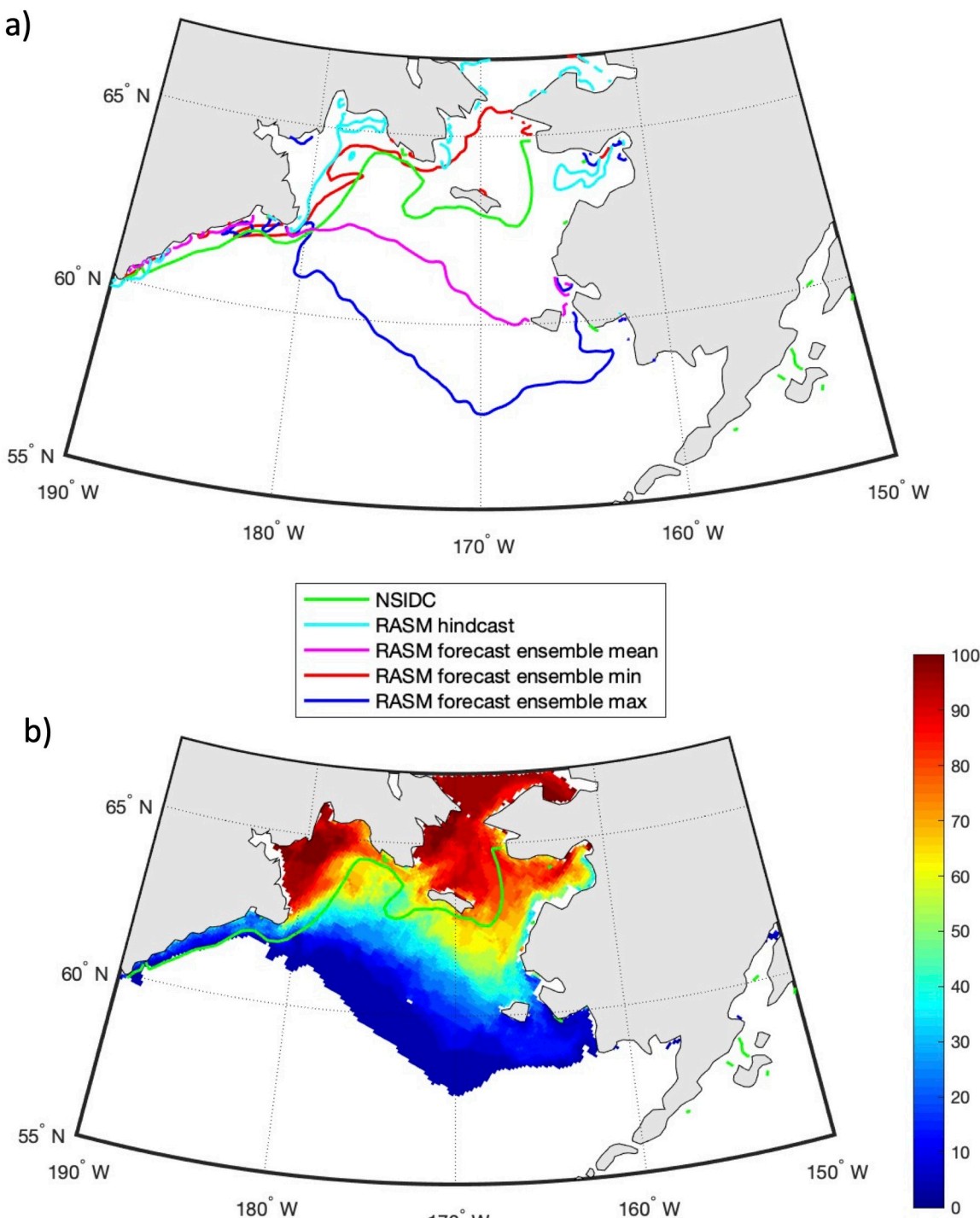

**Fig 12. Predictions of sea ice extent and probability for April 15, 2019.** (a) The RASM predicted ensemble mean (purple), maximum (blue) and minimum (red) sea ice extent on April 15, 2020, with all 31 individual member forecasts initialized on February 1, 2019. (b) The predicted probability of sea ice extent for April 15, 2019. The green line is the NSIDC observed sea ice extent [43] on April 15, 2019. The light blue line is the respective sea ice extent from the RASM hindcast. Figure created using Matlab (v. 2018a) from the authors' data.

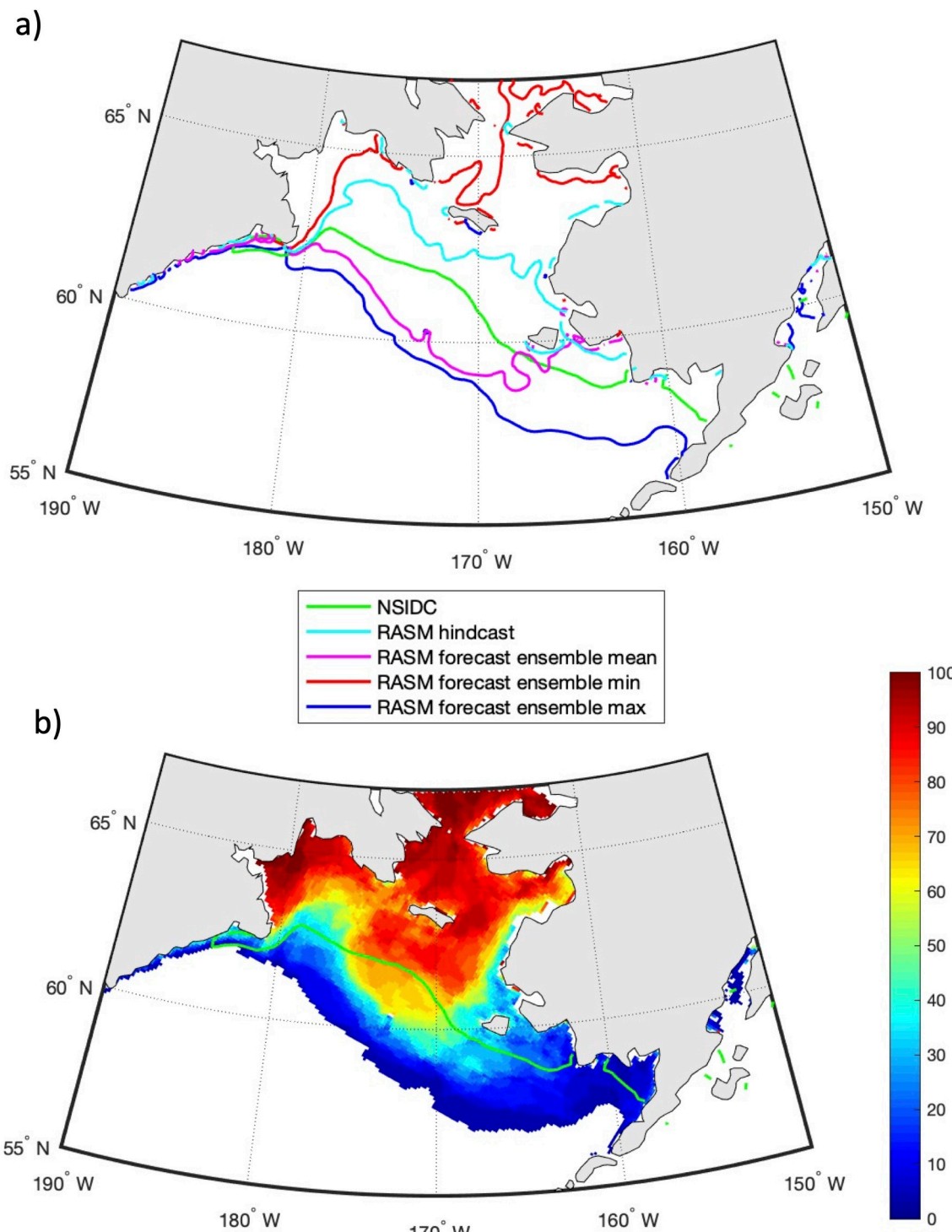

**Fig 13. Predictions of sea ice extent and probability for April 15, 2020.** (a) The RASM predicted ensemble mean (purple), maximum (blue) and minimum (red) sea ice extent on April 15, 2020, with all 31 individual member forecasts initialized on February 1, 2020. (b) The predicted probability of sea ice extent for April 15, 2020. The green line is the NSIDC observed sea ice extent [43] on April 15, 2020. The light blue line is the respective sea ice extent from the RASM hindcast. Figure created using Matlab (v. 2018a) from the authors' data.

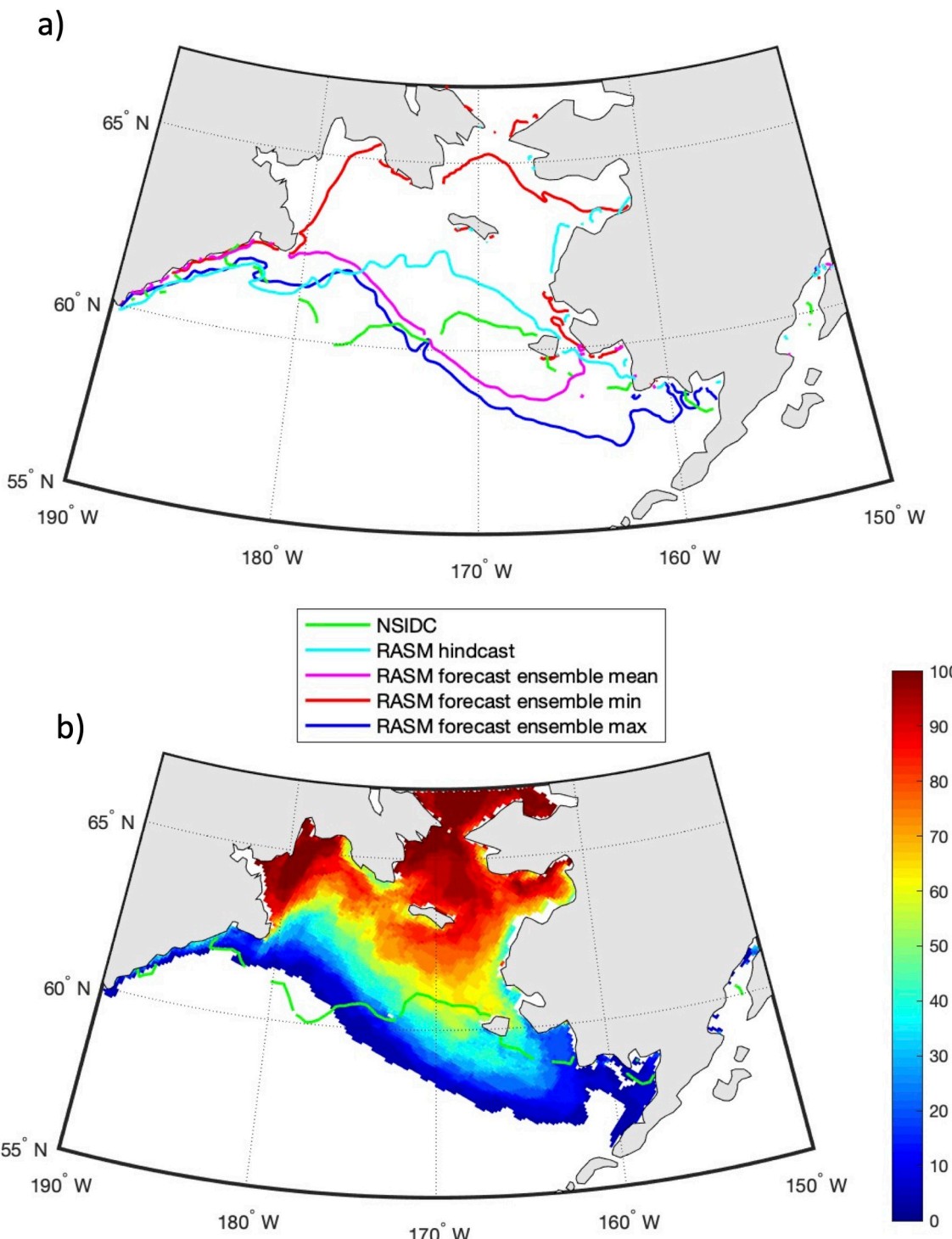

**Fig 14. Predictions of sea ice extent and probability for April 15, 2021.** (a) The RASM predicted ensemble mean (purple), maximum (blue) and minimum (red) sea ice extent on April 15, 2021, with all 31 individual member forecasts initialized on February 1, 2021. The green line is the NSIDC observed sea ice extent [44] on April 15, 2021. (b) The predicted probability of sea ice extent for April 15, 2021. Figure created using Matlab (v. 2018a) from the authors' data.

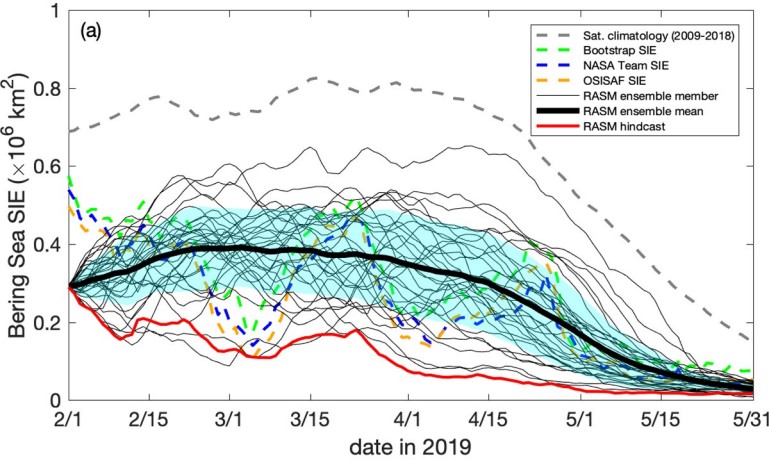

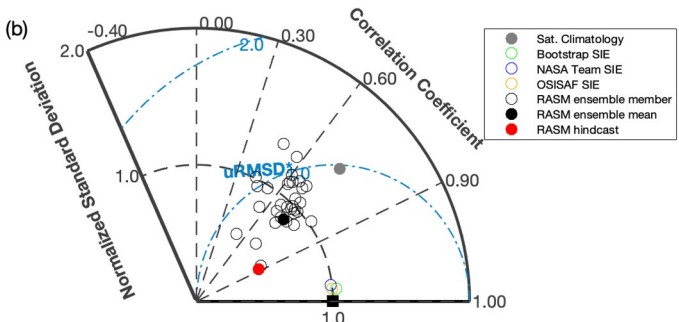

**Fig 15. Daily sea ice extent during February–May, 2019 from RASM compared with satellite observations.** (a) Daily timeseries of sea ice extent in the Bering Sea during February to May, 2019 from the RASM forecast, RASM hindcast, and satellite algorithms, with light blue shading representing 1 standard deviation from the forecast ensemble mean. The gray dashed line is the satellite-derived (Bootstrap, NASA NT, and OSISAF) daily climatology between 2009 and 2018. (b) Taylor diagram showing the normalized correlation coefficient (r), standard deviations (σ), and the unbiased root-mean-square difference (uRMSE) for the timeseries. The Taylor diagram illustrates a relative model performance in reproducing satellite-derived Bering Sea daily SIE variability, as well as the individual data sets, relative to the average of Bootstrap, NASA NT, and OSISAF products, represented by the black square as the reference where the normalized standard deviation and the correlation coefficient are 1.00 and the unbiased root mean difference is 0.00. Figure created using Matlab (v. 2018a).

The latest forecast is presented in Fig 14 for April 15, 2021 and shows an extent that is somewhat similar to 2020. However, the satellite observations show ice extending further south over deep water between 175˚W– 180˚. It is important to note that these figures show daily mean sea ice extent and are, therefore, highly influenced by short term atmospheric variability.

Figs 15–17 illustrate the 6-month ensemble forecasts of sea ice extent in Bering Sea using the RASM probabilistic prediction capability. Each ensemble was initialized on February 1st, 2019, 2020, and 2021, respectively. In 2019 and over the Bering Sea only, all the satellite data-sets (Bootstrap, NASA Team, and OSI SAF products) show lower ice cover but increased variability of sea ice extent (SIE) relative to the 2009–2018 satellite climatology data, likely due to variable wind forcing. SIE in the Bering Sea decreased through February and reached a winter minimum in March ($0.14 \pm 0.034 \times 10^6$ km$^2$; Fig 15A) before recovering back up to a maximum in late March of 2021. It was one of the lowest sea ice extents in the Bering Sea, which is

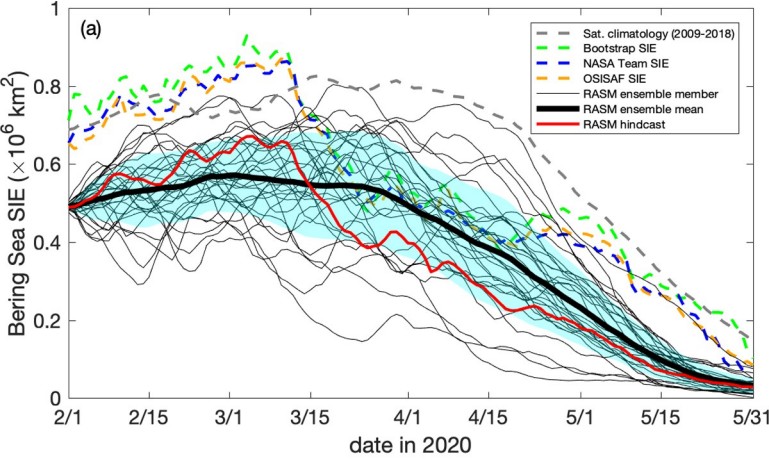

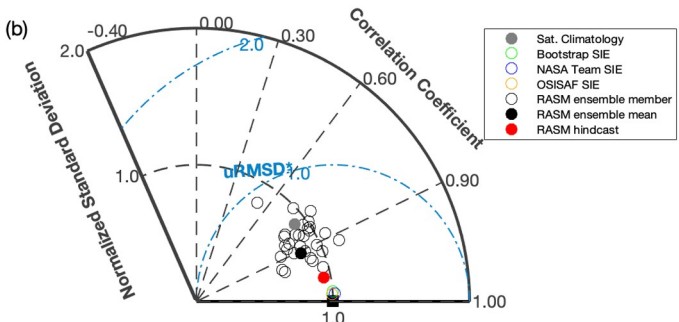

**Fig 16. Daily sea ice extent during February–May, 2020 from RASM compared with satellite observations.** (a) Daily timeseries of sea ice extent in the Bering Sea during February to May, 2020 from the RASM forecast, RASM hindcast, and satellite algorithms, with light blue shading representing 1 standard deviation from the forecast ensemble mean. The gray dashed line is the satellite-derived (Bootstrap, NASA NT, and OSISAF) daily climatology between 2009 and 2018. (b) Taylor diagram showing the normalized correlation coefficient (r), standard deviations (σ), and the unbiased root-mean-square difference (uRMSE) for the timeseries. The Taylor diagram illustrates a relative model performance in reproducing satellite-derived Bering Sea daily SIE variability, as well as the individual data sets, relative to the average of Bootstrap, NASA NT, and OSISAF products, represented by the black square as the reference where the normalized standard deviation and the correlation coefficient are 1.00 and the unbiased root mean difference is 0.00. Figure created using Matlab (v. 2018a).

captured well within the SIE spread of the 31-member ensemble, with the ensemble mean averaging out the temporal variability. The RASM SIE hindcast represents an underestimation, which suggests reduced accuracy of the prescribed atmospheric reanalysis, including near-surface winds. The RASM forecast skill, as measured by the SIE correlation coefficient (CC), was in the range between 0.55 and 0.75 for most ensemble members (74%) and the ensemble mean with their respective normalized standard deviation (SD) values close to those of the satellite SIE SD (Fig 15B). Compared to the satellite climatology, the skill of the ensemble mean forecast represents a gain. Based on these metrics, the RASM hindcast CC was higher (0.89) and its SD was lower (0.51) suggesting a very good agreement in terms of temporal variability but at a reduced magnitude.

In 2020 and 2021, SIE in the Bering Sea increased in February, reached the winter maximum in early March, and then gradually decreased in the following months, with the magnitudes comparable to the 2009–2018 SIE climatology (Figs 16A and 17A). Compared to

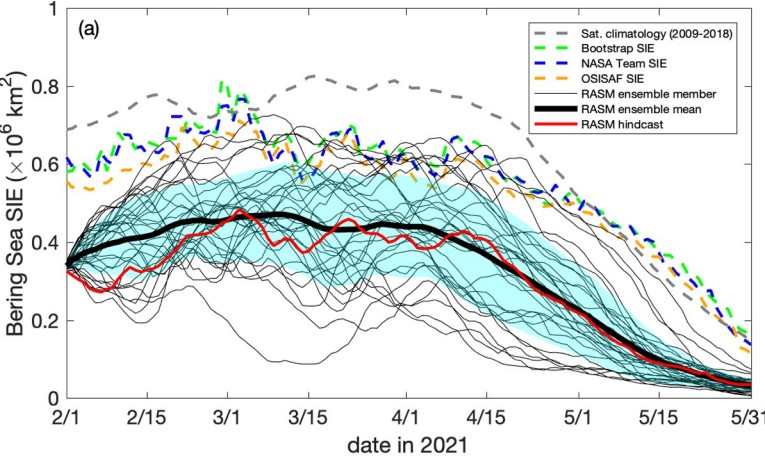

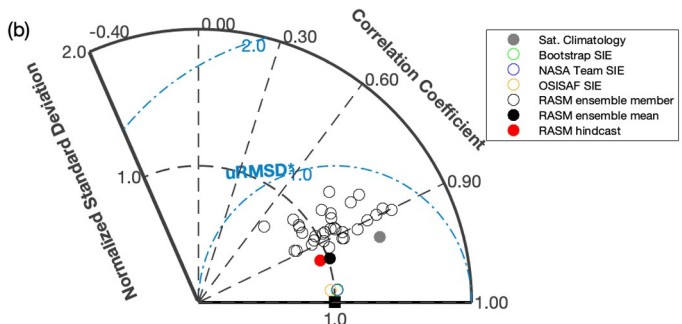

**Fig 17. Daily sea ice extent during February–May, 2021 from RASM compared with satellite observations.** (a) Daily timeseries of sea ice extent in the Bering Sea during February to May, 2021 from the RASM forecast, RASM hindcast, and satellite algorithms, with light blue shading representing 1 standard deviation from the forecast ensemble mean. The gray dashed line is the satellite-derived (Bootstrap, NASA NT, and OSISAF) daily climatology between 2009 and 2018. (b) Taylor diagram showing the normalized correlation coefficient (r), standard deviations (σ), and the unbiased root-mean-square difference (uRMSE) for the timeseries. The Taylor diagram illustrates a relative model performance in reproducing satellite-derived Bering Sea daily SIE variability, as well as the individual data sets, relative to the average of Bootstrap, NASA NT, and OSISAF products, represented by the black square as the reference where the normalized standard deviation and the correlation coefficient are 1.00 and the unbiased root mean difference is 0.00. Figure created using Matlab (v. 2018a).

observations, both the RASM ensemble mean forecast and the RASM hindcast captured the temporal variability but their SIEs were biased low, with only a few ensemble members showing comparable magnitudes. In terms of model forecast skill (Figs 16B and 17B), the ensemble mean CCs were significantly higher than in 2019, at 0.91 and 0.95, and their SDs closer to those in observations, at 0.84 and 1.0, in 2020 and 2021, respectively. While biased low in SIE these skills still represent a clear gain compared to the climatology. Overall, these analyses suggest a potential benefit of the RASM probabilistic forecast capability at time scales of up to 6 months, however it is uncertain how skillful these forecasts would be at longer time scales. Further improvements related to the magnitude of near surface wind speeds and their momentum coupling with the sea ice in the Bering Sea are the subjects of ongoing sensitivity studies.

We show in Fig 18 the predicted bottom water temperature for July 15, 2019, 2020 and 2021 from the same RASM forecasts described above. The predicted CP area of 2019 is the least of the 3 years with a value of 203,000 km$^2$, which is only 57% of the long-term mean CP

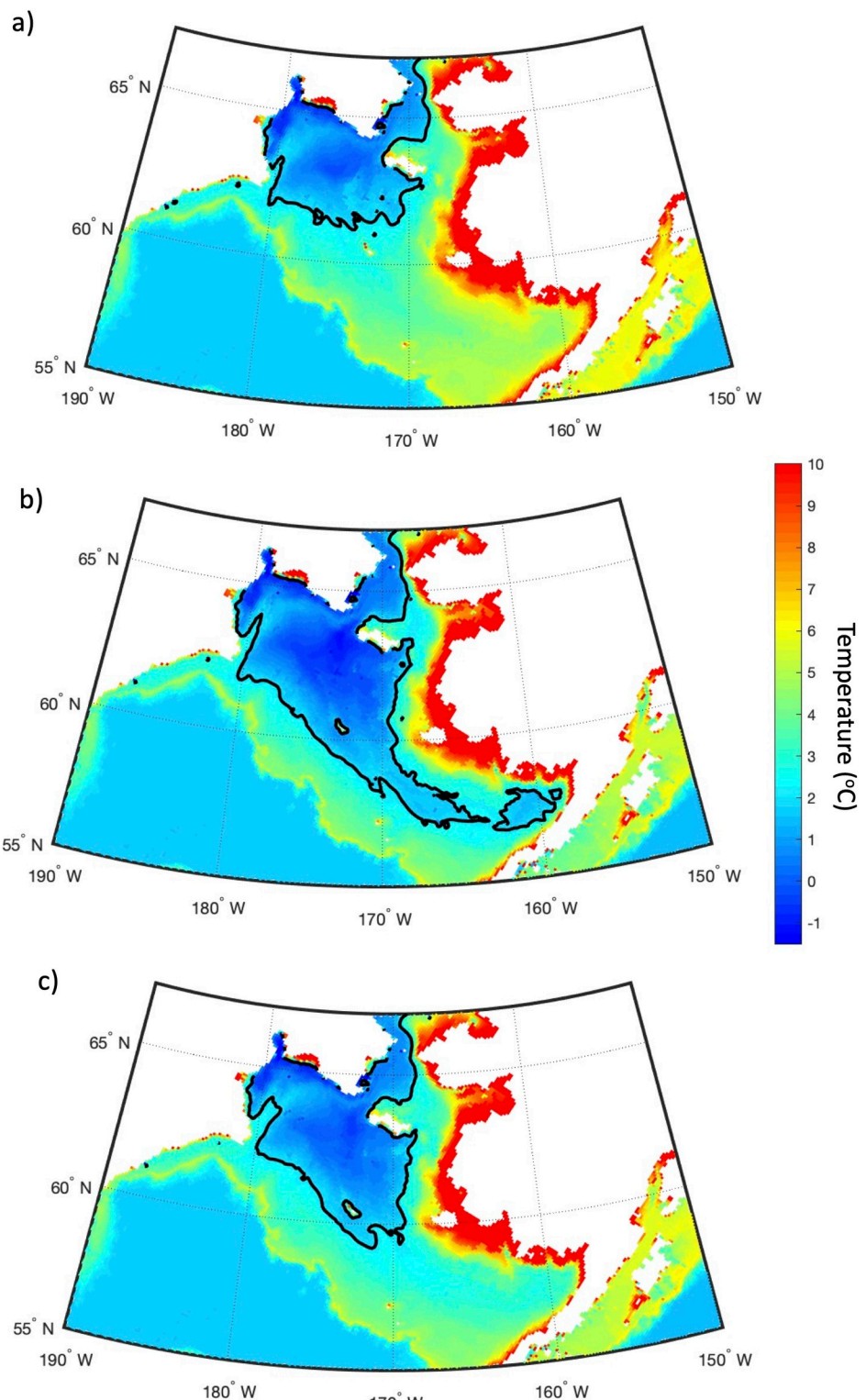

**Fig 18. Predicted bottom water temperature for July 15, 2019, 2020 and 2021.** Bottom water temperature (˚C) on July 15, 2019 (a), 2020 (b) and 2021 (c), as predicted by the RASM ensemble forecast initialized on February 1, 2019 (a), 2020 (b) and 2021 (c). The black contour line indicates the position of the 2˚C isotherm. Figure created using Matlab (v. 2018a) from the authors' data.

area during July. This predicted CP area for July 2019 is larger than the July 2018 minimum from the hindcast simulation (113,000 $km^2$). The CP of 2020 is the most extensive of the 3 years with an area of 368,000 $km^2$ within the 2˚C contour representing the predicted extent of the CP on July 15, 2020. This value is close to the long-term July mean of 359,000 $km^2$. The July 15, 2021 CP has an area of 265,000 $km^2$ in the RASM forecast, which is ~26% less than the long-term mean, but still over twice as large as the 2018 value. In a recent study by Kearney et al. [45] it was found that dynamic forecasting had some skill at predicting summer bottom temperatures across the eastern Bering Sea shelf with lead times of up to 4 months, however the skill level was lower when the forecast model was initialized during the fall or early winter. We conclude that while such predictions are now available, it is critical to collect measurements for their ongoing evaluation and potential tuning.

## Acknowledgments

We would like to thank an anonymous reviewer for helpful and insightful comments, which improved an earlier version of this manuscript. We also thank J. Grebmeier (University of Maryland Center for Environmental Science) for comments on the manuscript.

## Author Contributions

**Conceptualization:** Jaclyn Clement Kinney.

**Data curation:** Robert Osinski, Anthony Craig.

**Formal analysis:** Jaclyn Clement Kinney, Younjoo J. Lee, Karen Frey.

**Funding acquisition:** Wieslaw Maslowski.

**Investigation:** Jaclyn Clement Kinney.

**Methodology:** Jaclyn Clement Kinney, Robert Osinski, Anthony Craig.

**Resources:** Wieslaw Maslowski.

**Software:** Robert Osinski, Anthony Craig.

**Visualization:** Jaclyn Clement Kinney, Younjoo J. Lee, Karen Frey.

**Writing – original draft:** Jaclyn Clement Kinney, Wieslaw Maslowski, Younjoo J. Lee, Christina Goethel, Karen Frey.

**Writing – review & editing:** Jaclyn Clement Kinney, Younjoo J. Lee, Christina Goethel, Karen Frey.

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
