## [Decision Letter · Decision Letter 0]

28 Dec 2021

PONE-D-21-34414­­­On the variability of the Bering Sea Cold Pool and implications for the biophysical environmentPLOS ONE

Dear Dr. Clement Kinney,

Thank you for submitting your manuscript to PLOS ONE. After careful consideration, we feel that it has merit but does not fully meet PLOS ONE’s publication criteria as it currently stands. Therefore, we invite you to submit a revised version of the manuscript that addresses the points raised during the review process.

We look forward to receiving your revised manuscript.

Kind regards,

Xiaole Sun

Academic Editor

PLOS ONE

Journal Requirements:

2. We note that Figure(s) 1, 12, 13, 14, 18 and 19 in your submission contain map images which may be copyrighted. All PLOS content is published under the Creative Commons Attribution License (CC BY 4.0), which means that the manuscript, images, and Supporting Information files will be freely available online, and any third party is permitted to access, download, copy, distribute, and use these materials in any way, even commercially, with proper attribution. For these reasons, we cannot publish previously copyrighted maps or satellite images created using proprietary data, such as Google software (Google Maps, Street View, and Earth). For more information, see our copyright guidelines: http://journals.plos.org/plosone/s/licenses-and-copyright.

1. You may seek permission from the original copyright holder of Figure(s) 1, 12, 13, 14, 18 and 19  to publish the content specifically under the CC BY 4.0 license.  

Additional Editor Comments (if provided):

Both reviewers are positive to the manuscript that has been much improved. Please clarify the final few points before final publication.

Reviewers' comments:

Reviewer's Responses to Questions

**Comments to the Author**

1. Is the manuscript technically sound, and do the data support the conclusions?

Reviewer #1: Yes

Reviewer #2: Yes

2. Has the statistical analysis been performed appropriately and rigorously? 

Reviewer #1: Yes

Reviewer #2: Yes

3. Have the authors made all data underlying the findings in their manuscript fully available?

Reviewer #1: Yes

Reviewer #2: Yes

4. Is the manuscript presented in an intelligible fashion and written in standard English?

Reviewer #1: Yes

Reviewer #2: Yes

5. Review Comments to the Author

Reviewer #1: The authors have thoroughly addressed all of my previous comments and I think their work has really helped benefit this manuscript. My only remaining concern is just the short number of years used for the forecast comparisons. As previously stated, I think a much larger comparison is beyond the scope of this manuscript, which is OK, but I suggest that the authors add some short text in the discussion, clarifying the limited timespan during which conditions in the Bering Sea have also been anomalous. It’s entirely possible that the forecasts are doing well during these couple years, but may prove less skillful with a much larger sample size.

Reviewer #2: This contribution describes a set of downscaling simulations of the Bering Sea. The methods of analysis for the hindcasts and forecasts are generally sound (with the possible exception of Figures 15-18, see below), and the introductory and discussion materials are informative.

I have the following comments on the manuscript:

l. 97 "a long-term timeseries of the CP...has yet to be published" - this is not strictly true; the Kearney et al. (2020) paper cited in the references clearly includes such a timeseries (Fig. 5 of that work). Admittedly the present work does plot the entire year's results, whereas Kearney et al. focused on July conditions for their interannual timeseries. Other publications have shown long-term timeseries of bottom temperature, though not the cold pool per se.

l.106 -when first mentioned, it would be good to specify the exact domain of RASM (i.e. north of what latitude?), for those not familiar with this model.

l.127 "February 1, 2021" - do you mean February 1 of each forecast year?

Fig. 9 - why is a different palette used than for Fig. 6? Both illustrate ice concentration, so it would be helpful to use the same palette for both.

Fig. 10 - it would be helpful to add year/month labels to the four plots

l.377 - Bering Sea cold pool predictions. As of 2021, there is a published paper especially relevant to this section, which focuses on downscaled, CFS-driven forecasts of the Bering Sea. That recent paper may have been published *after* the present paper was composed, hence the authors would have been unaware of its existence:

Kearney, K. A., Alexander, M., Aydin, K., Cheng, W., Hermann, A. J., Hervieux, G., & Ortiz, I. (2021). Seasonal predictability of sea ice and bottom temperature across the eastern Bering Sea shelf. Journal of Geophysical Research: Oceans, 126, e2021JC017545. https://doi.org/10.1029/2021JC017545

In that paper, Kearney et al. quantify downscaled, monthly-averaged bottom temperature forecast skill, using ensembles initialized for each month of the year. Since the very same CFS global model output is used by both studies (in addition to two other global models used in Kearney), their published results would be highly relevant to the discussion section. In particular, they include forecast skills for March, April, and May, when the forecasts are initialized in February (the months used by this study).

l.436 (Figs 15-18). I assume that each day of each raw forecast time series is being used in the calculation of r, as opposed to each day's mean forecast value. Either way, it is at least slightly problematic to calculate r when that value is non-stationary. The correlation is naturally going to be high at the beginning of the forecasts, and then degrade over time - hence r is systematically non-stationary. It would be more "statistically correct" to calculate r separately for each forecast day or forecast month, despite the more limited sampling size available for this purpose. Having said this, it is perhaps no worse than calculating r between two independent spatial patterns (as is frequently done in oceanography), where one expects some areas to be consistently better correlated than others.

6. PLOS authors have the option to publish the peer review history of their article (what does this mean?). If published, this will include your full peer review and any attached files.

Reviewer #1: No

Reviewer #2: No

---

## [Author Response · Author response to Decision Letter 0]

9 Mar 2022

Reviewer #1: The authors have thoroughly addressed all of my previous comments and I think their work has really helped benefit this manuscript. My only remaining concern is just the short number of years used for the forecast comparisons. As previously stated, I think a much larger comparison is beyond the scope of this manuscript, which is OK, but I suggest that the authors add some short text in the discussion, clarifying the limited timespan during which conditions in the Bering Sea have also been anomalous. It’s entirely possible that the forecasts are doing well during these couple years, but may prove less skillful with a much larger sample size.

We have added this text to the Discussion “however it is uncertain how skillful these forecasts would be at longer time scales.”

Reviewer #2: This contribution describes a set of downscaling simulations of the Bering Sea. The methods of analysis for the hindcasts and forecasts are generally sound (with the possible exception of Figures 15-18, see below), and the introductory and discussion materials are informative.

I have the following comments on the manuscript:

l. 97 "a long-term timeseries of the CP...has yet to be published" - this is not strictly true; the Kearney et al. (2020) paper cited in the references clearly includes such a timeseries (Fig. 5 of that work). Admittedly the present work does plot the entire year's results, whereas Kearney et al. focused on July conditions for their interannual timeseries. Other publications have shown long-term timeseries of bottom temperature, though not the cold pool per se.

We have changed the wording:

“Near-bottom temperature observations on the southeastern Bering Sea shelf have noted changes in the distribution of the CP [e.g. [7,17]], especially in the central and eastern parts of the shelf. We present here a long-term timeseries of the full size of the CP, including its extension into Russian waters, utilizing results from the Regional Arctic System Model (RASM; [18]) and examine the linkage between CP area and sea ice area.”

l.106 -when first mentioned, it would be good to specify the exact domain of RASM (i.e. north of what latitude?), for those not familiar with this model.

We added this sentence: “This domain includes all oceanic areas of the northern hemisphere from 90oN to 55oN and most of the North Pacific down to 30oN.”

l.127 "February 1, 2021" - do you mean February 1 of each forecast year? 

Yes, thanks. Correction made.

Fig. 9 - why is a different palette used than for Fig. 6? Both illustrate ice concentration, so it would be helpful to use the same palette for both.

Unfortunately, Fig. 6 was created with different software than the other figures, so it cannot have the same palette.

Fig. 10 - it would be helpful to add year/month labels to the four plots

Years and months labels are present in the revised figure.

l.377 - Bering Sea cold pool predictions. As of 2021, there is a published paper especially relevant to this section, which focuses on downscaled, CFS-driven forecasts of the Bering Sea. That recent paper may have been published *after* the present paper was composed, hence the authors would have been unaware of its existence:

Kearney, K. A., Alexander, M., Aydin, K., Cheng, W., Hermann, A. J., Hervieux, G., & Ortiz, I. (2021). Seasonal predictability of sea ice and bottom temperature across the eastern Bering Sea shelf. Journal of Geophysical Research: Oceans, 126, e2021JC017545. https://doi.org/10.1029/2021JC017545

In that paper, Kearney et al. quantify downscaled, monthly-averaged bottom temperature forecast skill, using ensembles initialized for each month of the year. Since the very same CFS global model output is used by both studies (in addition to two other global models used in Kearney), their published results would be highly relevant to the discussion section. In particular, they include forecast skills for March, April, and May, when the forecasts are initialized in February (the months used by this study).

We have included your reference and text in the discussion.

l.436 (Figs 15-18). I assume that each day of each raw forecast time series is being used in the calculation of r, as opposed to each day's mean forecast value. Either way, it is at least slightly problematic to calculate r when that value is non-stationary. The correlation is naturally going to be high at the beginning of the forecasts, and then degrade over time - hence r is systematically non-stationary. It would be more "statistically correct" to calculate r separately for each forecast day or forecast month, despite the more limited sampling size available for this purpose. Having said this, it is perhaps no worse than calculating r between two independent spatial patterns (as is frequently done in oceanography), where one expects some areas to be consistently better correlated than others.

As you say, this is the way that R is frequently calculated in oceanography. We have done a very similar analysis (focused on the entire Arctic) recently in a paper by our group in the Journal of Climate (Watts et al. 2021). We believe this is the best way to present and statistically describe these results.

Watts, M., W. Maslowski, Y. J. Lee, J. Clement Kinney, R. Osinski (2021) A Spatial Evaluation of Arctic Sea Ice and Regional Limitations in CMIP6 Historical Simulations, Journal of Climate, https://doi.org/10.1175/JCLI-D-20-0491.1

---

## [Editor Report · Decision Letter 1]

16 Mar 2022

­­­On the variability of the Bering Sea Cold Pool and implications for the biophysical environment

PONE-D-21-34414R1

Dear Dr. Clement Kinney,

We’re pleased to inform you that your manuscript has been judged scientifically suitable for publication and will be formally accepted for publication once it meets all outstanding technical requirements.

Kind regards,

Xiaole Sun

Academic Editor

PLOS ONE

Additional Editor Comments (optional):

Thanks for taking care of all comments.
---

## [Editor Report · Acceptance letter]

17 Mar 2022

PONE-D-21-34414R1 

­­­On the variability of the Bering Sea Cold Pool and implications for the biophysical environment 

Dear Dr. Clement Kinney:

I'm pleased to inform you that your manuscript has been deemed suitable for publication in PLOS ONE. Congratulations! Your manuscript is now with our production department. 

Kind regards, 

on behalf of

Dr. Xiaole Sun 

Academic Editor

PLOS ONE